# Development of the Italian Clinical Practice Guidelines on Bariatric and Metabolic Surgery: Design and Methodological Aspects

**DOI:** 10.3390/nu15010189

**Published:** 2022-12-30

**Authors:** Maurizio De Luca, Marco Antonio Zappa, Monica Zese, Ugo Bardi, Maria Grazia Carbonelli, Francesco Maria Carrano, Giovanni Casella, Marco Chianelli, Sonja Chiappetta, Angelo Iossa, Alessandro Martinino, Fausta Micanti, Giuseppe Navarra, Giacomo Piatto, Marco Raffaelli, Eugenia Romano, Simone Rugolotto, Roberto Serra, Emanuele Soricelli, Antonio Vitiello, Luigi Schiavo, Iris Caterina Maria Zani, Giulia Bandini, Edoardo Mannucci, Benedetta Ragghianti, Matteo Monami

**Affiliations:** 1Rovigo Hospital, 45030 Rovigo, Italy; 2ASST Fatebenefratelli-Sacco, 20157 Milan, Italy; 3Casa di Cura Privata Salus, 84091 Salerno, Italy; 4A.O. San Camillo Forlanini, 00152 Rome, Italy; 5Minimally Invasive Unit, Department of Surgery, Università Degli Studi di Roma “Tor Vergata”, 00173 Rome, Italy; 6Department of Surgical Sciences, Sapienza University of Rome, AOU Policlinico Umberto I, 00161 Rome, Italy; 7Ospedale Regina Apostolorum Roma, 00041 Rome, Italy; 8Ospedale Evangelico Betania Napoli, 80147 Naples, Italy; 9Sapienza Polo Pontino Dipartimento di Scienze Biotecnologie Medico Chirurgiche, 04100 Latina, Italy; 10Division of Transplantation, Department of Surgery, University of Illinois at Chicago, Chicago, IL 60607, USA; 11Università Federico II Napoli, 80138 Naples, Italy; 12Policlinico Universitario “G. Martino” Messina, 98124 Messina, Italy; 13Ospedale di Montebelluna, 31044 Montebelluna, Italy; 14Fondazione Policlinico Universitario Agostino Gemelli IRCCS, Università Cattolica del Sacro Cuore Roma, 00168 Rome, Italy; 15Department of Psychological Medicine, Institute of Psychiatry, Psychology & Neuroscience, King’s College London, London WC2R 2LS, UK; 16Policlinico Casa di Cura Abano Terme, 35031 Abano Terme, Italy; 17Ospedale Santa Maria Nuova Firenze, 50122 Florence, Italy; 18Department of Advanced Biomedical Sciences, Università Degli Studi Di Napoli “Federico II”, 80138 Naples, Italy; 19Department of Medicine and Surgery, University of Salerno, Baronissi, 84084 Fisciano, Italy; 20Amici Obesi ONLUS Milano, 20128 Milan, Italy; 21Diabetes Agency, Azienda Ospedaliero Universitaria Careggi and University of Florence, 50134 Florence, Italy

**Keywords:** obesity, metabolic/bariatric surgery (MBS), type 2 diabetes (T2D), arterial hypertension (AHI), dyslipidemia (DL), obstructive sleep apnea (OSA), gastroesophageal reflux disease (GERD), guidelines

## Abstract

Development of the Italian clinical practice guidelines on bariatric and metabolic surgery, as well as design and methodological aspects. Background: Obesity and its complications are a growing problem in many countries. Italian Society of Bariatric and Metabolic Surgery for Obesity (Società Italiana di Chirurgia dell’Obesità e delle Malattie Metaboliche—SICOB) developed the first Italian guidelines for the treatment of obesity. Methods: The creation of SICOB Guidelines is based on an extended work made by a panel of 24 members and a coordinator. Grading of Recommendations, Assessment, Development and Evaluations (GRADE) methodology has been used to decide the aims, reference population, and target health professionals. Clinical questions have been created using the PICO (Patient, Intervention, Comparison, Outcome) conceptual framework. The definition of questions used the two-step web-based Delphi method, made by repeated rounds of questionnaires and a consensus opinion from the panel. Results: The panel proposed 37 questions. A consensus was immediately reached for 33 (89.2%), with 31 approved, two rejected and three which did not reach an immediate consensus. The further discussion allowed a consensus with one approved and two rejected. Conclusions: The areas covered by the clinical questions included indications of metabolic/bariatric surgery, types of surgery, and surgical management. The choice of a surgical or a non-surgical approach has been debated for the determination of the therapeutic strategy and the correct indications.

## 1. Introduction

Obesity and its complications are a growing public health concern in many countries, due an increasing prevalence, a relevant impact on the health of affected individuals and a growing related economic burden [1]. Treatments for obesity, which include lifestyle interventions and drug therapies, are often characterized by limited long-term efficacy [2]. Metabolic surgery, which has been developed for achieving a relevant weight loss in morbidly obese individuals, has also be shown to have a therapeutic potential for obesity-related conditions, such as type 2 diabetes (T2D) [3,4] and obstructive sleep apnea [5]. However, the use of surgical approaches has been limited by organizational and economic limitations.

The development of rigorous guidelines is a relevant tool for the improvement of the quality of care, increasing the appropriateness of therapeutic choices. The Italian Society of Bariatric and Metabolic Surgery for Obesity (Società Italiana di Chirurgia dell’Obesità e delle Malattie Metaboliche—SICOB) recognized this need and decided to design and develop the first Italian guidelines, aimed at assisting healthcare professionals in the consideration of the surgical option for the treatment of obesity and related conditions. In the Italian national legal environment [6], the inclusion of guidelines in the National Guideline System is possible only after a careful methodological and formal revision by the National Center for Clinical Excellence of the Ministry of Health. In the development of national guidelines, the Center for Clinical Excellence recommends the use of Grading of Recommendations, Assessment, Development and Evaluations (GRADE) methodology [7], which requires the identification of specific clinical questions and the definition of relevant outcomes for each one of those questions. The present paper reports the development of questions, the definition of outcomes for each question, and the description of search strategy and study inclusion criteria for each outcome.

## 2. Materials and Methods

### 2.1. Characteristics of the Panel Involved in the Development of the Guideline

Panel members, identified by SICOB in collaboration with other scientific societies Appendix A, elected a coordinator and nominated the members of the evidence review team. The latter actively collects and analyses evidence, without participating in the definition of clinical questions, outcomes, and recommendations. The complete list of the 24 members of the panel, with their roles, and 10 from the evidence review team, is reported in Appendix A. The mean age of panelists was 48.0 ± 11.9 years, with 46.1% aged less than 45 years.

All members of the panel and evidence review team compiled a declaration of potential conflicts of interest, which were collectively discussed to determine their relevance. In all cases, the reported conflicts were considered minimal or irrelevant; therefore, all components of the panel and of the evidence review team participated in the elaboration of all recommendations.

### 2.2. GRADE Methodology for the Development of Guidelines

The GRADE method [8] was developed to reduce the impact of personal opinions and prejudices on the recommendations of guidelines, inducing a greater adherence to evidence derived from methodologically valid studies. The first step of the development of guidelines, following this method, is the definition of a scoping document, defining aims, reference population, and target health professionals.

The following step is that of defining a series of clinical questions, using the PICO (Patient, Intervention, Comparison, Outcome) conceptual framework [7]. Each recommendation is developed as the answer to a question.

For each question, the panel defines a number of clinical outcomes, which are potentially relevant for the choice of different clinical options. Each outcome is then rated (from 1 to 9) for its importance; those receiving a rating of 7 or higher are classified as “critical” and represent the basis for the development of the recommendation.

For each critical outcome, the evidence review team will perform a systematic review of relevant studies, predefining search strategies and inclusion criteria, and performing meta-analyses whenever possible Table 1 and Appendix A. Studies and related meta-analyses are assessed for methodological quality in order to verify the actual strength of available evidence.

Further assessments include economic evaluations (usually based on cost-utility ratio, whenever possible), organizational impact, equity, acceptability, and feasibility. The final recommendation includes all those elements.

### 2.3. Delphi Process

The definition of questions was performed using a two-step web-based Delphi method, a structured technique aimed at obtaining, by repeated rounds of questionnaires, a consensus opinion from a panel of experts in areas wherein evidence is scarce or conflicting, and opinion is important [9].

Between June and October 2022, panelists were invited to propose questions with the PICO framework and to express their level of agreement or disagreement on each proposed question using a 5-point Likert scale, scored from 1 to 5 (1, strongly disagree; 2, disagree; 3, agree; 4, mostly agree; and 5, strongly agree). Results were expressed as a percentage of respondents who scored each item as 1 or 2 (disagreement) or as 3, 4, or 5 (agreement). A positive consensus was reached in case of more than 66% agreement, a negative consensus in case of more than 66% disagreement, consensus was not reached when the sum for disagreement or agreement was below 66% [9]. For the statements on which consensus had not been achieved, panelists were asked to re-rate in a second round their agreement/disagreement, after internal discussion with all panelists.

## 3. Results

These guidelines will apply to adolescents (age > 13 years), adults, patients with Body Mass Index (BMI) > 30 kg/m^2^ requiring bariatric or metabolic surgery. Healthcare systems, infrastructures, human and financial resources across Italian regions will be considered in developing these guidelines. Therefore, they are primarily intended to be applicable in Italy. The present guidelines will be used by healthcare professionals, including surgeons, obesity specialists, general practitioners, nutrition experts, psychologists, endocrinologists/diabetologists, and pediatricians.

### Clinical Questions

The panel therefore identified 32 clinical questions, organized into six domains:

A.Indication for surgery (11 questions);B.Perioperative work-up/management (9 questions);C.Bariatric procedures (5 questions);D.Endoscopic procedures (1 question);E.Revisional surgery (2 questions);F.Postoperative care (4 questions).

The approved questions and their related approved critical outcomes are reported in Table 1. Proposed outcomes not reaching the rating for being considered critical are reported in Appendix A.

The evidence review team identified the characteristics of relevant studies for each critical outcome, defining search strategy and study inclusion criteria, which are reported in Appendix A. The search strategy used for all questions is: “obesity AND surgery” with an expected start date on 1 December 2022.

## 4. Discussion

The areas covered by the clinical questions identified by panelists include indications of metabolic surgery, types of surgery, and pre-, peri- and post-surgical management of obese patients. The focus on indications is not surprising: the choice of a surgical or a non-surgical approach for the treatment of obesity and related metabolic conditions is often debated, posing relevant issues for the determination of appropriateness of the therapeutic strategy. In addition, considering the current legislation on professional liability [6], a correct identification of proper indications can support clinicians in an environment characterized by increasing legal claims. In patients referred to surgical treatment, the choice of the most appropriate intervention is a major concern for surgeons; the collection and synthesis of available evidence from methodologically valid studies can be a more appropriate support for this decision than personal beliefs or experience.

The focus on procedures to be applied before, during and after the surgical procedure, is less obvious. Interestingly, the identification of those questions was performed by a panel mainly composed of surgeons, showing a clinical vision much broader than the surgical act per se.

Some questions were devoted to the application of surgical procedure for the treatment of obesity-related conditions, such as T2D. The possibility of using surgical techniques to treat T2D and other diseases associated with excess weight, even in patients with mild obesity, or not even properly obese, has been pursued by some authors [3,4]; promising preliminary evidences suggest that bariatric surgery has metabolic effects beyond weight loss, justifying the name of “metabolic surgery” [10]. However, the use of surgery in patients with relatively low BMI, even though affected by concomitant conditions, is still debated [11], and it is not recommended by most guidelines [12]. The systematic collection of evidence will depict a clearer picture of the effects of surgery in these conditions, thus completing existing guidelines on medical treatments for T2D [13,14] and other diseases.

Obese patients seeking treatment have expectations of weight loss. Conversely, health professionals have a greater attention to obesity-associated metabolic abnormalities, such as hyperglycemia, dyslipidemia, etc. The panelist planning the development of these guidelines reached one step further, recognizing the central role of longer-term hard outcomes, such as mortality, incident cardiovascular disease, and malignancies. The availability of sufficient evidence for a reliable assessment of the effects of surgery on those outcomes will be verified in the process of developing these guidelines. Appropriately sized, long-term studies on hard outcomes can be considered a priority for research.

The choice of a specific therapeutic strategy should be based on the assessment of the risk-benefit ratio, together with cost-utility analysis. This means that adverse events need to be systematically and carefully studied. In fact, safety outcomes have been included for most clinical questions, concurring to the development of recommendations.

Transparency in the development process is one of the main determinants of quality of guidelines [15,16,17,18,19] Potential conflicts of interest and some explanations on the data underlying recommendations are provided by most guidelines. The GRADE manual recommends the explicit publication of clinical questions, relevant outcomes, and summaries of evidence for each outcome [8]. We decided to go beyond the requirements of the GRADE manual, pre-emptively publishing in extenso the whole process leading to clinical questions and definition of critical outcomes. In addition, the search strategy and inclusion criteria for the systematic review and meta-analysis for each outcome has been reported in the present study, allowing the reproducibility of the whole process. It is the policy of this panel to publish extensively, and possibly on peer-reviewed journals, all systematic reviews and meta-analyses that will concur to the formulation of these guidelines.

## 5. Conclusions

The creation of SICOB Guidelines is based on an extended work. Grading of Recommendations, Assessment, Development and Evaluations (GRADE) methodology has been used to decide aims, reference population, and target health professionals. Clinical questions have been created using PICO (Patient, Intervention, Comparison, Outcome) conceptual framework. The GRADE manual recommends the explicit publication of clinical questions, relevant outcomes, and summaries of evidence for each outcome. The search strategy and inclusion criteria for the systematic review and meta-analysis for each outcome has been reported in the present study, allowing for the reproducibility of the whole process.

## Figures and Tables

**Table 1 nutrients-15-00189-t001:** Delphi survey results and outcomes approval process.

**N**	**PICO**	**Disagreement** **(Score 1–2)**	**Agreement** **(Score 3–5)**	**Outcome** **(Median)**	**Approval**
	A. Indication for surgery				
1	In patients with uncontrolled type 2 diabetes and BMI 30–34.9 kg/m^2^, is bariatric/metabolic surgery preferable to non-bariatric and metabolic surgical treatments, for the treatment of diabetes?	4.2%	95.8%	-	
	Outcomes (efficacy)				
1.1	Diabetes remission			8	
1.2	Improvement of glycometabolic control (HbA1c; FPG; lipid profile; blood pressure)	8	
1.3	Decrease of body weight (BMI; percentage of weigh lost; percentage of fat mass)	8	
1.4	Reduction of macrovascular complications			8	
1.5	Reduction of all-cause mortality			8	
1.6	Improvement of quality of life			8	
	Outcomes (safety)				
1.7	Perioperative mortality			7	
1.8	Perioperative surgical complications			7	
1.9	Serious adverse events (surgical and non-surgical)			7	
2	In patients with uncontrolled type 2 diabetes and BMI ≥ 35 kg/m^2^, is bariatric and metabolic surgery preferable to non-bariatric and metabolic surgical treatments, for the treatment of diabetes?	0%	100%	-	
	Outcomes (efficacy)				
2.1	Diabetes remission			8	
2.2	Improvement of glycometabolic control (HbA1c; FPG; lipid profile; blood pressure)	8	
2.3	Decrease of body weight (BMI; percentage of weigh lost; percentage of fat mass)	8	
2.4	Reduction of macrovascular complications			8	
2.5	Reduction of all-cause mortality			8	
2.6	Improvement of quality of life			8	
	Outcomes (safety)				
2.7	Perioperative mortality			7	
2.8	Perioperative surgical complications			7	
2.9	Serious adverse events (surgical and non-surgical)			7	
3	In patients with BMI 30–34.9 kg/m^2^ and at least one uncontrolled comorbid condition (diabetes, hypertension, dyslipidemia, obstructive sleep apnea), is bariatric and metabolic surgery preferable to non-bariatric and metabolic surgical treatments, for the treatment of obesity?	0%	100%	-	
	Outcomes (efficacy)				
3.1	Diabetes remission			8	
3.2	Improvement of glycometabolic control (HbA1c; FPG; lipid profile; blood pressure)	8	
3.3	Decrease of body weight (BMI; percentage of weigh lost; percentage of fat mass)	8	
3.4	Reduction of macrovascular complications			8	
3.5	Reduction of all-cause mortality			8	
3.6	Improvement of quality of life			8	
	Outcomes (safety)				
3.7	Perioperative mortality			8	
3.8	Perioperative surgical complications			7	
3.9	Serious adverse events (surgical and non-surgical)			7	
4	In patients with BMI ≥ 35 kg/m^2^ and at least one comorbid condition (diabetes, hypertension, dyslipidemia, obstructive sleep apnea), is bariatric and metabolic surgery preferable to non-bariatric and metabolic surgical treatments, for the treatment of obesity?	0%	100%	-	
	Outcomes (efficacy)				
4.1	Diabetes remission			8	
4.2	Improvement of glycometabolic control (HbA1c; FPG; lipid profile; blood pressure)	8	
4.3	Decrease of body weight (BMI; percentage of weigh lost; percentage of fat mass)	8	
4.4	Reduction of macrovascular complications			8	
4.5	Reduction of all-cause mortality			8	
4.6	Improvement of quality of life			8	
4.7	Hypertension remission	8	
4.8	Obesity-related complication remission	8	
	Outcomes (safety)				
4.9	Perioperative mortality			8	
4.10	Perioperative surgical complications			7.5	
4.11	Serious adverse events (surgical and non-surgical)			7.5	
5	In patients with BMI ≥ 40 kg/m^2^, is bariatric and metabolic surgery preferable to non-bariatric and metabolic surgical treatments, for the treatment of obesity?	0%	100%	-	
	Outcomes (efficacy)				
5.1	Diabetes remission			8	
5.2	Improvement of glycometabolic control (HbA1c; FPG; lipid profile; blood pressure)	8	
5.3	Decrease of body weight (BMI; percentage of weigh lost; percentage of fat mass)	9	
5.4	Reduction of macrovascular complications			8.5	
5.5	Reduction of all-cause mortality			8.5	
5.6	Improvement of quality of life			8	
5.7	Hypertension remission	8	
5.8	Obesity-related complication remission	8	
	Outcomes (safety)				
5.9	Perioperative mortality			7.5	
5.10	Perioperative surgical complications			7.5	
5.11	Serious adverse events (surgical and non-surgical)			8	
6	In pediatric patients with BMI ≥ 30 kg/m^2^, is bariatric/metabolic surgery preferable to non-bariatric/metabolic surgical treatments, for the treatment of obesity?	16.7%	83.3%	-	
	Outcomes (efficacy)				
6.1	Obesity-related complication remission	8	
6.2	Decrease of body weight (BMI; percentage of weigh lost; percentage of fat mass)	8	
6.3	Reduction of all-cause mortality			7	
6.4	Improvement of quality of life			8	
	Outcomes (safety)				
6.5	Perioperative mortality			8	
6.6	Perioperative surgical complications			7.5	
6.7	Serious adverse events (surgical and non-surgical)			7	
7	In patients with BMI ≥ 30 kg/m^2^ and age > 60years, is bariatric/metabolic surgery preferable to non-bariatric/metabolic surgical treatments, for the treatment of obesity?	0%	100%	-	
	Outcomes (efficacy)				
7.1	Obesity-related complication remission	8	
7.2	Decrease of body weight (BMI; percentage of weigh lost; percentage of fat mass)	8	
7.3	Reduction of all-cause mortality			8	
7.4	Improvement of quality of life			8	
	Outcomes (safety)				
7.5	Perioperative mortality			7.5	
7.6	Perioperative surgical complications			8	
7.7	Serious adverse events (surgical and non-surgical)			8	
8	In patients with BMI ≥ 30 kg/m^2^ and gastroesophageal reflux disease (GERD), is bariatric/metabolic surgery preferable to non-bariatric/metabolic surgical treatments, for the treatment of GERD?	0%	100%	-	
	Outcomes (efficacy)				
8.1	Reduction of the incidence of Barrett disease	8	
8.2	Reduction of the incidence of gastro-esophageal malignancies	7	
8.3	Decrease of body weight (BMI; percentage of weigh lost; percentage of fat mass)	7	
8.4	Improvement of quality of life			8	
	Outcomes (safety)				
8.5	Perioperative surgical complications			6	
8.6	Serious adverse events (surgical and non-surgical)			6	
9	In patients with BMI ≥ 30 kg/m^2^ and arthropathy, is bariatric/metabolic surgery preferable to non-bariatric/metabolic surgical treatments, for the treatment of arthropathy?	0%	100%	-	
	Outcomes (efficacy)				
9.1	Reduction of hospital stay	7	
9.2	Reduction of all-cause mortality	7.5	
9.3	Decrease of body weight (BMI; percentage of weigh lost; percentage of fat mass)	8	
9.4	Reduction of re-hospitalization	7	
9.5	Reduction of perioperative orthopedic surgical complications	8	
9.6	Improvement of quality of life			8	
	Outcomes (safety)				
9.7	Perioperative surgical (bariatric) complications			7	
9.8	Perioperative mortality			7	
9.9	Serious adverse events (surgical and non-surgical)			7	
10	In patients with BMI ≥ 30 kg/m^2^ with indication for renal/hepatic transplantation, is bariatric/metabolic surgery preferable to non-bariatric/metabolic surgical treatments, for increasing the eligibility for renal/hepatic transplantation?	4.2%	95.8%	-	
	Outcomes (efficacy)				
10.1	Increase of transplantation eligibility	8	
10.2	Reduction of surgical (transplantation) complications	8	
10.3	Decrease of graft rejection	8	
	Outcomes (safety)				
10.4	Perioperative surgical (bariatric) complications			7	
10.5	Serious adverse events (surgical and non-surgical)			7	
11	In patients with BMI ≥ 30 kg/m^2^, is bariatric/metabolic surgery preferable to non-bariatric/metabolic surgical treatments, for preventing incident malignancies?	4.2%	95.8%	-	
	Outcomes (efficacy)				
11.1	Reduction of incident malignancies	8	
11.2	Reduction of mortality for cancer	8	
	Outcomes (safety)				
11.3	Perioperative surgical (bariatric) complications			6.5	
11.4	Serious adverse events (surgical and non-surgical)			5	
	B. Peri-operative work-up/management				
12	In patients with BMI ≥ 30 kg/m^2^ with indication to bariatric/metabolic surgery, the pre-operative screening of obstructive sleep apnea is preferable to non-screening, for reducing peri-operative complications?	4.2%	95.8%	-	
	Outcomes (efficacy)				
12.1	Improvement of apnea-hypopnea index			8	
12.2	Reduction of perioperative mortality			8	
12.3	Increase of undiagnosed apnea detection	6	
12.4	Decrease of body weight (BMI; percentage of weigh lost; percentage of fat mass)	6.5	
	*Outcomes* (safety)				
12.5	Perioperative surgical complications			7	
12.6	Length of hospitalization			6	
13	In patients with BMI ≥ 30 kg/m^2^ with indication to bariatric/metabolic surgery and obstructive sleep apnea, the peri-operative use of Continuous Positive Airway Pressure (C-PAP) is preferable to non-using C-PAP, for reducing peri-operative complications?	0%	100%	-	
	Outcomes (efficacy)				
13.1	Improvement of apnea-hypopnea index			8	
13.2	Decrease of perioperative surgical complications	8	
13.3	Reduction of perioperative mortality			8	
13.4	Detection of patients with undiagnosed apnea	6.5	
	Outcomes (safety)				
13.5	Reduced compliance/acceptability			6.5	
14	In patients with BMI ≥ 30 kg/m^2^ with indication to bariatric/metabolic surgery, is a pre-operative gastroscopy preferable to non-performing a pre-operative gastroscopy, for reducing peri-operative complications?	4.2%	95.8%	-	
	Outcomes (efficacy)				
14.1	Reduction of surgical dehiscence			7	
14.2	Reduction of re-intervention	7	
14.3	Reduction of all-cause mortality	7	
	Outcomes (safety)				
14.4	Perioperative surgical complications			7	
14.5	Length of surgical procedure			5	
14.6	Length of hospitalization			5	
15	In patients with BMI ≥ 30 kg/m^2^ with indication to bariatric/metabolic surgery, the pre-operative weight loss is preferable to non-weight loss, for reducing peri-operative complications?	12.5%	87.5%	-	
	Outcomes (efficacy)				
15.1	Reduction of peri-operative surgical complications			8	
15.2	Reduction of length of surgical procedures	7.5	
15.3	Decrease of body weight (BMI; percentage of weigh lost; percentage of fat mass)	7.5	
15.4	Improvement of quality of life			8	
	Outcomes (safety)				
15.5	Increase of time-to-surgery			6.5	
16	In patients with BMI ≥ 30 kg/m^2^ with indication to bariatric/metabolic surgery, the peri-operative use of anticoagulants, is preferable to non-using anticoagulants, for reducing peri-operative thromboembolic complications?	0%	100%	-	
	Outcomes (efficacy)				
16.1	Reduction of peri-operative mortality	8	
16.2	Reduction of surgical complications	8	
16.3	Reduction of thromboembolic complications	9	
16.4	Reduction of hospital stay			6.5	
	Outcomes (safety)				
16.5	Increase of bleeding			8	
16.6	Increase of thrombocytopenia			6	
17	In patients with BMI ≥ 30 kg/m^2^ with indication to bariatric/metabolic surgery, the peri-operative use of antibiotic thearpy, is preferable to non-using antibiotic therapy, for reducing peri-operative infective complications?	12.5%	87.5%	-	
	Outcomes (efficacy)				
17.1	Reduction of peri-operative infective complications			8	
17.2	Reduction of peri-operative mortality	7	
17.3	Reduction of peri-operative surgical complications	7	
17.4	Reduction of hospital stay			7	
	Outcomes (safety)				
17.5	Increase of creatinine levels			5.5	
17.6	Increase of incident renal failure			5	
18	In patients with BMI ≥ 30 kg/m^2^ with indication to bariatric/metabolic surgery, the peri-operative use of Enhanced Recovery After Bariatric Surgery (ERABS) protocols, is preferable to non-using ERABS protocols, for increasing post-operative functional recovery?	4.2%	95.8%	-	
	Outcomes (efficacy)				
18.1	Reduction of peri-operative surgical complications			8	
18.2	Reduction of time to patient mobilization	8	
18.3	Reduction of post-surgical pain	8	
18.4	Reduction of hospital stay			8	
18.5	Reduction of time for enteral feeding/hydration			7	
18.6	Reduction of all-cause mortality	8	
18.7	Increase of quality of life			8	
18.8	Increase of number of surgical procedures	6.5	
	*Outcomes* (safety)				
18.9	Increase of re-hospitalization			6.5	
19	In patients with BMI ≥ 30 kg/m^2^ with indication to bariatric/metabolic surgery, the peri-operative use of vitamin D (and other vitamins/calcium) supplementation, is preferable to non-using supplementation, for preventing/treating vitamin deficiency?	4.2%	95.8%	-	
	Outcomes (efficacy)				
19.1	Increase of 25-OH vitamin D serum levels	8	
19.2	Increase of other vitamins and total protein serum levels	6.5	
19.3	Decrease of body weight (BMI; percentage of weigh lost; percentage of fat mass)	6	
	Outcomes (safety)				
19.4	Increase of serum calcium levels			6	
19.5	Increase of incident renal failure			5	
19.6	Increase of transaminase levels			5	
20	In patients with BMI ≥ 30 kg/m^2^ with indication to bariatric/metabolic surgery, the peri-operative use of ursodeoxycholic acid therapy, is preferable to non-using ursodeoxycholic acid therapy, for preventing gallbladder stones?	0%	100%	-	
	Outcomes (efficacy)				
20.1	Reduction of incident gallbladder stones	8	
20.2	Reduction of cholecystectomy	7	
	Outcomes (safety)				
20.3	Increase of surgical complications			6	
**N**	**PICO**	**Disagreement** **(Score 1–2)**	**Agreement** **(Score 3–5)**	**Outcome** **(Median)**	**Approval**
	C. Bariatric procedures				
21	In patients with uncontrolled type 2 diabetes and BMI 30–34.9 kg/m^2^, which type of bariatric and metabolic surgery is preferable for the treatment of diabetes?	16.7%	83.3%	-	
	Outcomes (efficacy)				
21.1	Diabetes remission			8	
21.2	Improvement of glycometabolic control (HbA1c; FPG; lipid profile; blood pressure)	8	
21.3	Decrease of body weight (BMI; percentage of weigh lost; percentage of fat mass)	7	
21.4	Reduction of macrovascular complications			8	
21.5	Reduction of all-cause mortality			7.5	
21.6	Improvement of quality of life			7	
	Outcomes (safety)				
21.7	Perioperative mortality			7	
21.8	Perioperative surgical complications			7.5	
21.9	Serious adverse events (surgical and non-surgical)			7	
22	In patients with uncontrolled type 2 diabetes and BMI ≥ 35 kg/m^2^, which type of bariatric and metabolic surgery is preferable, for the treatment of diabetes?	4.2%	95.8%	-	
	Outcomes (efficacy)				
22.1	Diabetes remission			8	
22.2	Improvement of glycometabolic control (HbA1c; FPG; lipid profile; blood pressure)	8	
22.3	Decrease of body weight (BMI; percentage of weigh lost; percentage of fat mass)	8	
22.4	Reduction of macrovascular complications			8	
22.5	Reduction of all-cause mortality			8	
22.6	Improvement of quality of life			8	
	Outcomes (safety)				
22.7	Perioperative mortality			7	
22.8	Perioperative surgical complications			7	
22.9	Serious adverse events (surgical and non-surgical)			7	
23	In patients with BMI 30–34.9 kg/m^2^ and at least one uncontrolled comorbid condition (diabetes, hypertension, dyslipidemia, obstructive sleep apnea), which type of bariatric and metabolic surgery is preferable, for the treatment of obesity?	8.3%	91.7%	-	
	Outcomes (efficacy)				
23.1	Diabetes remission			8	
23.2	Improvement of glycometabolic control (HbA1c; FPG; lipid profile; blood pressure)	8	
23.3	Decrease of body weight (BMI; percentage of weigh lost; percentage of fat mass)	8	
23.4	Reduction of macrovascular complications			8	
23.5	Reduction of all-cause mortality			7	
23.6	Improvement of quality of life			8	
	Outcomes (safety)				
23.7	Perioperative mortality			8	
23.8	Perioperative surgical complications			8	
23.9	Serious adverse events (surgical and non-surgical)			7.5	
24	In patients with BMI ≥ 35 kg/m^2^ and at least one comorbid condition (diabetes, hypertension, dyslipidemia, obstructive sleep apnea), which type of bariatric and metabolic surgery is preferable, for the treatment of obesity?	8.3%	91.7%	-	
	Outcomes (efficacy)				
24.1	Diabetes remission			8	
24.2	Improvement of glycometabolic control (HbA1c; FPG; lipid profile; blood pressure)	8	
24.3	Decrease of body weight (BMI; percentage of weigh lost; percentage of fat mass)	8	
24.4	Reduction of macrovascular complications			8	
24.5	Reduction of all-cause mortality			8	
24.6	Improvement of quality of life			8	
24.7	Hypertension remission	8	
24.8	Metabolic complications remission	8	
	Outcomes (safety)				
24.9	Perioperative mortality			8	
24.10	Perioperative surgical complications			8	
24.11	Serious adverse events (surgical and non-surgical)			7.5	
25	In patients with BMI ≥ 40 kg/m^2^, which type of bariatric/metabolic surgery is preferable, for the treatment of obesity?	8.3%	91.7%	-	
	Outcomes (efficacy)				
25.1	Diabetes remission			8	
25.2	Improvement of glycometabolic control (HbA1c; FPG; lipid profile; blood pressure)	8	
25.3	Decrease of body weight (BMI; percentage of weigh lost; percentage of fat mass)	9	
25.4	Reduction of macrovascular complications			8	
25.5	Reduction of all-cause mortality			8	
25.6	Improvement of quality of life			8	
25.7	Hypertension remission	8	
25.8	Metabolic complications remission	8	
	Outcomes (safety)				
25.9	Perioperative mortality			8	
25.10	Perioperative surgical complications			7.5	
25.11	Serious adverse events (surgical and non-surgical)			7	
**N**	**PICO**	**Disagreement** **(Score 1–2)**	**Agreement** **(Score 3–5)**	**Outcome** **(Median)**	**Approval**
	D. Endoscopic procedures				
26	In patients with BMI ≥ 30 kg/m^2^, is primary endoscopic surgical treatment preferable to non-endoscopic surgical treatment, for the treatment of obesity?	12.5%	87.5%	-	
	Outcomes (efficacy)				
26.1	Diabetes remission	7	
26.2	Improvement of glycometabolic control (HbA1c; FPG; lipid profile; blood pressure)	7	
26.3	Decrease of body weight (BMI; percentage of weigh lost; percentage of fat mass)	8	
26.4	Reduction of macrovascular complications			7	
26.5	Reduction of all-cause mortality			7	
26.6	Improvement of quality of life			8	
	*Outcomes* (safety)				
26.7	Perioperative mortality			7.5	
26.8	Perioperative surgical complications			7	
26.9	Serious adverse events (surgical and non-surgical)			7.5	
**N**	**PICO**	**Disagreement** **(Score 1–2)**	**Agreement** **(Score 3–5)**	**Outcome** **(Median)**	**Approval**
	E. Revisional surgery				
27	In patients with BMI ≥ 30 kg/m^2^, who underwent bariatric/metabolic surgery with weight regain, is a new surgical treatment preferable to non-surgical treatment, for treating weight regain?	4.2%	95.8%	-	
	Outcomes (efficacy)				
27.1	Prevention of incidence/recurrence of diabetes			7	
27.2	Improvement of glycometabolic control (HbA1c; FPG; lipid profile; blood pressure)	7	
27.3	Decrease of body weight (BMI; percentage of weigh lost; percentage of fat mass)	7.5	
27.4	Reduction of macrovascular complications			7	
27.5	Reduction of all-cause mortality			7	
27.6	Improvement of quality of life			7	
	Outcomes (safety)				
27.7	Perioperative mortality			7	
27.8	Perioperative surgical complications			7	
27.9	Serious adverse events (surgical and non-surgical)			7	
28	In patients with BMI ≥ 30 kg/m^2^, who underwent bariatric/metabolic surgery and weight regain, is a new surgical treatment preferable to medical therapy with drugs approved for the treatment of obesity, for treating weight regain?	4.2%	95.8%	-	
	Outcomes (efficacy)				
28.1	Prevention of incidence/recurrence of diabetes			7	
28.2	Improvement of glycometabolic control (HbA1c; FPG; lipid profile; blood pressure)	7	
28.3	Decrease of body weight (BMI; percentage of weigh lost; percentage of fat mass)	7	
28.4	Reduction of macrovascular complications			7	
28.5	Reduction of all-cause mortality			8	
28.6	Improvement of quality of life			8	
	Outcomes (safety)				
28.7	Perioperative mortality			8	
28.8	Perioperative surgical complications			7	
28.9	Serious adverse events (surgical and non-surgical)			7	
**N**	**PICO**	**Disagreement** **(Score 1–2)**	**Agreement** **(Score 3–5)**	**Outcome** **(Median)**	**Approval**
	F. Post-operative care				
29	In patients with BMI ≥ 30 kg/m^2^, who underwent bariatric/metabolic surgery, is medical therapy with drugs approved for the treatment of obesity preferable to non-pharmacological treatment, for maintaining weight loss?	4.2%	95.8%	-	
	Outcomes (efficacy)				
29.1	Prevention of incidence/recurrence of diabetes			7	
29.2	Improvement of glycometabolic control (HbA1c; FPG; lipid profile; blood pressure)	7	
29.3	Decrease of body weight (BMI; percentage of weigh lost; percentage of fat mass)	7	
29.4	Reduction of macrovascular complications			7	
29.5	Reduction of all-cause mortality			7	
29.6	Improvement of quality of life			7	
	Outcomes (safety)				
29.7	Perioperative mortality			7	
29.8	Perioperative surgical complications			7	
29.9	Serious adverse events (surgical and non-surgical)			7	
30	In patients with BMI ≥ 30 kg/m^2^, who underwent bariatric/metabolic surgery, is post-surgical multidisciplinary follow-up preferable to non-adopting multidisciplinary follow-up, for maintaining weight loss?	4.2%	95.8%	-	
	Outcomes (efficacy)				
30.1	Prevention of incidence/recurrence of diabetes			7	
30.2	Improvement of glycometabolic control (HbA1c; FPG; lipid profile; blood pressure)	7	
30.3	Decrease of body weight (BMI; percentage of weigh lost; percentage of fat mass)	9	
30.4	Reduction of weight regain	8	
30.5	Improvement of quality of life			8.5	
30.6	Reduction of depressive symptoms			6	
	Outcomes (safety)				
30.7	Reduction of compliance to educational programs			6	
31	In patients with BMI ≥ 30 kg/m^2^, who underwent bariatric/metabolic surgery, is life-style modification programs preferable to non-adopting life-style modification programs, for maintaining weight loss?	0%	100%	-	
	Outcomes (efficacy)				
31.1	Prevention of incidence/recurrence of diabetes			7	
31.2	Improvement of glycometabolic control (HbA1c; FPG; lipid profile; blood pressure)	7	
31.3	Decrease of body weight (BMI; percentage of weigh lost; percentage of fat mass)	9	
31.4	Reduction of weight regain	8	
31.5	Improvement of quality of life			9	
31.6	Reduction of depressive symptoms			6	
	Outcomes (safety)				
31.7	Increase of alcohol or other substances abuse			6.5	
32	In patients with BMI ≥ 30 kg/m^2^, who underwent bariatric/metabolic surgery, is planning pregnancy after weight loss stabilization preferable to planning pregnancy during weight loss, for preventing maternal-fetal adverse events?	0%	100%	-	
	Outcomes (efficacy)				
32.1	Reduction of cesarean delivery			7.5	
32.2	Reduction of pre-term delivery	8	
32.3	Reduction of post-partum hemorrhage	6.5	
	Outcomes (safety)				
32.4	Increase of weight gain during pregnancy			6	
32.5	Increase of sideropenic anemia			6.5	

List of abbreviations: HbA1c: glicated Hemoglobin; FPG: fasting plasma glucose test; Increase of 25-OH vitamin D serum levels: with the use of single vitamin D.

## Data Availability

Not applicable.

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
