# Peer review of "Development of the Italian Clinical Practice Guidelines on Bariatric and Metabolic Surgery: Design and Methodological Aspects"

_nutrients, 2022, doi:10.3390/nu15010189_

Round 1

Reviewer 1 Report

Dear Authors,

This is a very interesting article and an excellent initiative. The topic is modern and we need guidelines on such an important issue as it is obesity and bariatric/metabolic surgery to specifically address this matter.

I have a few observations:

1.    The title is not included in the main abstract.

In site – the title is “Development of the Italian clinical practice guidelines on bariatric and metabolic surgery: design and methodological aspects

This title suggests several papers are coming/expected in order to finally address the actual clinical practice guidelines. If this is the first step, I suggest to exclude the part of “design and methodological aspects” since “development” includes Methods. Also, “design” is part of “methodology”.

2.    No need for point after authors’ list and keywords’ list

3.    I suggest “bariatric surgery” as well as part of the keywords since this is a very used term

4.    Introduction – Line 59 -  typo “Countries” with capital letter

5.    Introduction – Line 74 – typo “environment6”

6.    Methods – Table S1 and Lines 88-89 - It is not clear if the 24 members of the panel are the entire panel since “panel” is separately mentioned after authors’ list (after the actual list of names)? Who are the members of “evidence review team for…”?

7.    GRADE section – Lines 114-116 introduce the methodology underlying the outcomes with a reference to Table S2. However, the Table S2 introduces “PICO not reaching consensus and requiring a second Delphi round”

8.    Table S2. Please define the yellow and red signs. Why some words are underlined? Why you choose Italics for some lines? Please introduce the abbreviations when first used (“BMI” which is later explained in line 144). Please, explain at least initially what do you mean by “diabetes” – type 2 diabetes mellitus, other types of diabetes, and different criteria of glucose profile anomalies including impaired glucose tolerance, etc.? The same observations goes for “hypertension” – meaning “arterial hypertension” which should be mentioned at least once since there are many other types of hypertension

9.    Lines 163 - 164 – “defining search strategy and study inclusion criteria, which are reported in Table S2”.  There is no such information in Table S2

10. Table 1 - Lines named 1.2, 1.3. 2.2, 2.3., 3.2., 3.3., 4.2., 4.3., 5.2., 5.3., 6.2., 7.2., 8.3., 9.3. are steeping over the columns dedicated to scores

11. Table 1 - Section 19. There is a major difference concerning the use of vitamin D with calcium and single use of vitamin D. Also, “increase of 25-OH vitamin D levels” means normalization or correction of deficiency since there is a different approach? A list of abbreviations after the table is needed. (“FPG”, “HBA1c”, etc.)

12. The references concerning Table S3 should be placed at the end of the paper

Thank you,

Best regards

Author Response

.    The title is not included in the main abstract. Done

In site – the title is “Development of the Italian clinical practice guidelines on bariatric and metabolic surgery: design and methodological aspects”

This title suggests several papers are coming/expected in order to finally address the actual clinical practice guidelines. If this is the first step, I suggest to exclude the part of “design and methodological aspects” since “development” includes Methods. Also, “design” is part of “methodology”. This is a critical point. We discussed regarding this item. We think that the partial elimination of the introduction is a distortion that not allows the title to have its complete explanation.

  1. No need for point after authors’ list and keywords’ list Done
  2. I suggest “bariatric surgery” as well as part of the keywords since this is a very used term Done, but Bariatric/metabolic surgery is a more used word
  3. Introduction – Line 59 -  typo “Countries” with capital letter  Done
  4. Introduction – Line 74 – typo “environment6”  Done
  5. Methods – Table S1 and Lines 88-89 - It is not clear if the 24 members of the panel are the entire panel since “panel” is separately mentioned after authors’ list (after the actual list of names)? Who are the members of “evidence review team for…”? There are other members who are part of these guidelines but they are not part of this panel, they are authors or members of the evidence review team.
  6. GRADE section – Lines 114-116 introduce the methodology underlying the outcomes with a reference to Table S2. However, the Table S2 introduces “PICO not reaching consensus and requiring a second Delphi round” DONE, I insert Table 1-S2 at the right position in the text.
  7. Table S2. Please define the yellow and red signs. Why some words are underlined? Why you choose Italics for some lines? Please introduce the abbreviations when first used (“BMI” which is later explained in line 144). Please, explain at least initially what do you mean by “diabetes” – type 2 diabetes mellitus, other types of diabetes, and different criteria of glucose profile anomalies including impaired glucose tolerance, etc.? The same observations goes for “hypertension” – meaning “arterial hypertension” which should be mentioned at least once since there are many other types of hypertension Thank you. Done
  8. Lines 163 - 164 – “defining search strategy and study inclusion criteria, which are reported in Table S2”.  There is no such information in Table S2I insert Table 1-S2 at the right position in the text.
  9. Table 1 - Lines named 1.2, 1.3. 2.2, 2.3., 3.2., 3.3., 4.2., 4.3., 5.2., 5.3., 6.2., 7.2., 8.3., 9.3. are steeping over the columns dedicated to scores DONE
  10. Table 1 - Section 19. There is a major difference concerning the use of vitamin D with calcium and single use of vitamin D. Also, “increase of 25-OH vitamin D levels” means normalization or correction of deficiency since there is a different approach? A list of abbreviations after the table is needed. (“FPG”, “HBA1c”, etc.) DONE. It is with the single use of Vit D.
  11. The references concerning Table S3 should be placed at the end of the paper DONE

Thank you,

Best regards

Reviewer 2 Report

In the paper, the authors try to propose a guideline for the treatment of obesity using the GRADE methodology and PICO framework, which is an interesting and critical question. However, there are some concerns.

1.     Line 1: there is no title for the article.

2.     Line 8-9: authors' institutes should line up.

3.     Line 40: Countries should be countries.

4.     Line 47: reference format is not consistent with the rest of the paper, and the referenced paper didn’t mention PICO at all, need to use the correct reference.

5.     Line 49: The author should make sure the total questions proposed, there are 6 in Table S2 and 32 questions in Table 1, so a total of 38 questions is proposed.

6.     Line 51: Conclusions should be capitalized.

7.     Line 55, 56: keywords are not accurate, and need to redefine.

8.     Lines 59-60: has format and grammar mistakes.

9.     Line 74: didn’t find the organization “Italian national legal environmet6”.

10.  Line 88-89: is the 10 evidence review team included in the 24-panel members? If not, they are not included in Table S1.

11.  Line 108: again, reference 7 didn’t mention PICO, need the correct reference.

12.  Line 116: Table S2 should be the definition of search strategies, and inclusion criteria, but not disagreed clinical questions.

13.  Line 119: Table S2, the format of kg/m2 format should be consistent; 6 should have a star for the second round of votes.

14.  Line 143: These guidelines refer to what? The first sentence doesn’t need to be indented to be consistent with other sessions.

15.  Line 161: Table S3 content is not what is described in the text. Each paragraph should be double line spaced.

16.  Line 164: Table S2 content is not correct; “obesity and surgery”.

17.  Line 187: questions 21-25 for type C are asking “which type of bariatric and metabolic surgery is preferable for the treatment of diabetes” which is not an agree or disagree question. Authors should explain how it can be used for consensus polls.

18.  Line 198: under D type questions, question 1 should be 31.

19.  Line 200: Table S3: didn’t find the content of this table discussed in the results. Authors should clarify or make a supplement in the results; RCT refers to?

20.  Lines 224 to 234: should combine with the reference at the end.

21.  Line 341 to 349: Authors should keep the accuracy of the reference list, didn’t see 16-19 cited anywhere.

Author Response

  1. Line 1: there is no title for the article. DONE
  2. Line 8-9: authors' institutes should line up.DONE
  3. Line 40: Countries should be countries. DONE
  4.    Line 47: reference format is not consistent with the rest of the paper, and the referenced paper didn’t mention PICO at all, need to use the correct reference. Thank you. The reference is correct because describes GRADE method and in the text the PICO
  5. Line 49: The author should make sure the total questions proposed, there are 6 in Table S2 and 32 questions in Table 1, so a total of 38 questions is proposed. Others are been refused. Yes, the others have been refused.
  6. Line 51: Conclusions should be capitalized. DONE
  7. Line 55, 56: keywords are not accurate, and need to redefine. DONE
  8. Lines 59-60: has format and grammar mistakes. DONE
  9. Line 74: didn’t find the organization “Italian national legal environmet6”. DONE
  10. Line 88-89: is the 10 evidence review team included in the 24-panel members? If not, they are not included in Table S1. There are other members who are part of these guidelines but they are not part of this panel
  11. Line 108: again, reference 7 didn’t mention PICO, need the correct reference. Thank you. The reference is correct because describes GRADE method and in the text the PICO
  12. Line 116: Table S2 should be the definition of search strategies, and inclusion criteria, but not disagreed clinical questions. Yes, they are reported
  13. Line 119: Table S2, the format of kg/m2format should be consistent; 6 should have a star for the second round of votes.
  14. Line 143: These guidelines refer to what? The first sentence doesn’t need to be indented to be consistent with other sessions. Italian clinical practice guidelines on bariatric and metabolic surgery Done
  15. Line 161: Table S3 content is not what is described in the text. Each paragraph should be double line spaced. DONE
  16. Line 164: Table S2 content is not correct; “obesity and surgery”. Yes, it is correct.
  17. Line 187: questions 21-25 for type C are asking “which type of bariatric and metabolic surgery is preferable for the treatment of diabetes” which is not an agree or disagree question. Authors should explain how it can be used for consensus polls. We have not the answer at this stage. Probably the severe methodology of GRADE doesn’t give us the possibility to answer to your right and fundamental question.
  18. Line 198: under D type questions, question 1 should be 31. DONE
  19. Line 200: Table S3: didn’t find the content of this table discussed in the results. Authors should clarify or make a supplement in the results; RCT refers to? Done
  20. Lines 224 to 234: should combine with the reference at the end. DONE
  21. Line 341 to 349: Authors should keep the accuracy of the reference list, didn’t see 16-19 cited anywhere. DONE

Thank you.

Best regards.